# Genomic Characterization of *Aeromonas veronii* Provides Insights into Taxonomic Assignment and Reveals Widespread Virulence and Resistance Genes throughout the World

**DOI:** 10.3390/antibiotics12061039

**Published:** 2023-06-11

**Authors:** José Cleves da Silva Maia, Gabriel Amorim de Albuquerque Silva, Letícia Stheffany de Barros Cunha, Gisele Veneroni Gouveia, Aristóteles Góes-Neto, Bertram Brenig, Fabrício Almeida Araújo, Flávia Aburjaile, Rommel Thiago Jucá Ramos, Siomar Castro Soares, Vasco Ariston de Carvalho Azevedo, Mateus Matiuzzi da Costa, João José de Simoni Gouveia

**Affiliations:** 1Graduate Program in Animal Science, Agricultural Sciences Campus, Federal University of Vale of São Francisco (Univasf), Petrolina 56304-917, Pernambuco, Brazil; cleves.maia@discente.univasf.edu.br (J.C.d.S.M.);; 2Center for Open Access Genomic Analysis (CALAnGO), Federal University of Vale of São Francisco (Univasf), Petrolina 56304-917, Pernambuco, Brazil; 3Laboratory of Molecular Computational Biology of Fungi (LBMCF), Institute of Biological Sciences, Federal University of Minas Gerais, Belo Horizonte 31270-901, Minas Gerais, Brazil; 4Institute of Veterinary Medicine, University of Göttingen, 37077 Göttingen, Niedersachsen, Germany; 5Biological Engineering Laboratory, Institute of Biological Sciences, Federal University of Pará (UFPA), Belém 66075-110, Pará, Brazil; 6Preventive Veterinary Medicine Department, Veterinary School, Federal University of Minas Gerais, Belo Horizonte 31270-901, Minas Gerais, Brazil; 7Department of Microbiology, Immunology, and Parasitology, Federal University of Triângulo Mineiro, Uberaba 38025-180, Minas Gerais, Brazil; 8Laboratory of Cellular and Molecular Genetics (LGCM), Institute of Biological Sciences, Federal University of Minas Gerais, Belo Horizonte 31270-901, Minas Gerais, Brazil

**Keywords:** *Aeromonas*, antibiotic resistance, misclassification, genomics, virulence, colistin resistance, One Health

## Abstract

*Aeromonas veronii* is a Gram-negative bacterial species that causes disease in fish and is nowadays increasingly recurrent in enteric infections of humans. This study was performed to characterize newly sequenced isolates by comparing them with complete genomes deposited at the NCBI (National Center for Biotechnology Information). Nine isolates from fish, environments, and humans from the São Francisco Valley (Petrolina, Pernambuco, Brazil) were sequenced and compared with complete genomes available in public databases to gain insight into taxonomic assignment and to better understand virulence and resistance profiles of this species within the One Health context. One local genome and four NCBI genomes were misidentified as *A. veronii*. A total of 239 virulence genes were identified in the local genomes, with most encoding adhesion, motility, and secretion systems. In total, 60 genes involved with resistance to 22 classes of antibiotics were identified in the genomes, including *mcr-7* and *cphA*. The results suggest that the use of methods such as ANI is essential to avoid misclassification of the genomes. The virulence content of *A. veronii* from local isolates is similar to those complete genomes deposited at the NCBI. Genes encoding colistin resistance are widespread in the species, requiring greater attention for surveillance systems.

## 1. Introduction

The genus *Aeromonas* comprises more than 30 species of Gram-negative bacteria commonly associated with the aquatic environment and is known to infect and cause mortality in fish [1,2]. Among the most important species of the genus is *Aeromonas veronii*, a mobile mesophilic bacterium that causes furunculosis and motile aeromonas septicemia (MAS) in fish, affecting commercial fish farms and causing economic losses to aquaculture [3,4]. In addition, the role of this species as a pathogen capable of causing intestinal diseases in humans has been increasingly highlighted [5,6].

Several components contribute to the pathogenicity of *A. veronii*, from lateral and polar flagella, more than one subtype of type IV pili and non-filamentous adhesins, to secretion systems that release toxins and exoenzymes into the environment or deliver directly into target cells [7,8]. These virulence factors help the pathogen during the processes of colonization, adhesion, and destruction of host cells, thus overcoming the immune system [2]. Among them, type I and type II secretion systems (*T1SS* and *T2SS*), type IV pili (*T4P*), and polar flagellum are conserved within the species; whereas type III, type V, and type VI secretion systems (*T3SS*, *T5SS*, and *T6SS*) show variable dispersion among isolates [8]. In addition, important virulent strains may have more than one copy of *T3SS*, as well as some toxins and iron uptake systems [9].

Species of the genus *Aeromonas* are already naturally resistant to some beta-lactams [1]. In the *A. veronii* resistome, antibiotic resistance genes (ARGs) can be grouped according to the mechanism of resistance into at least four categories: antibiotic efflux, antibiotic inactivation, antibiotic target alteration, and antibiotic target replacement [8]. Through horizontal gene transfer (HGT), these bacteria could acquire and disseminate ARGs, especially in the aquatic environment, from where multidrug-resistant strains may emerge [10,11,12]. This situation may involve resistance to antibiotics that are considered the last line of defense against resistant Gram-negative bacteria, such as resistance to colistin encoded by *mcr* (mobile colistin resistance) genes, since these are increasingly disseminated in the aquatic environment, with probable participation from *Aeromonas* spp. in the process [10,13,14,15].

Although it is known as one of the main pathogens responsible for cases of MAS in fish [4], *A*. *veronii* is amongst the main *Aeromonas* species associated with infections in humans, being isolated mainly from feces, blood, and wounds [2]. In some cases, this species is the main bacteria isolated from patients with gastroenteritis, and the frequency of infections increases in warmer seasons and in older hosts [5]. In this sense, population aging and the ubiquity of *Aeromonas* spp. may facilitate a future scenario in which infections by these bacteria become a major health problem [2].

As an emerging pathogen in humans [2,5,6,16], and as likely sources for ARGs [12,13,14], investigating the epidemiology of *A. veronii* becomes increasingly important; however, care must be taken when identifying this and other species of the genus isolated from animals, humans, and the environment, since classification errors in *Aeromonas* spp. are common when using phenotypic methods [2,17,18,19]. Regarding this, genome sequencing and the use of in silico tools to classify species are fundamental [17,18], considering that this also enables for the characterization of virulence profile and allows for comparative analyses with more virulent strains isolated from different hosts, leading to the identification of essential components behind the pathogenicity [6,8,9]. Another essential analysis is to annotate and identify ARGs from local isolates and compare them with those present in genomes from other countries to monitor the global spread of resistance to important antibiotics, acting under the context of a One Health approach [20].

Taking into account the importance of genomic sequencing to characterize, classify, and monitor the resistance and virulence of pathogens, this study aims to perform genomic analyses of bacterial isolates from the São Francisco Valley (Petrolina, Pernambuco, Brazil) classified as *A. veronii* and from complete genomes available in public databases to gain insight into the taxonomic assignment of the species *A. veronii* and to better understand the virulence and resistance profiles of this species in context of One Health.

## 2. Results

### 2.1. MLST, TETRA, ANI, and Phylogeny

MLST typing (Multi-Locus Sequence Typing) reveals that all genomes belong to the genus *Aeromonas*, although only seven NCBI complete genomes have had their sequence type recognized (485, 333, 166, 515, 1000, 160, and 254). All sets of alleles from the complete genomes differ at least in one of the six genes used in typing (*gyrB*, *groL*, *gltA*, *metG*, *ppsA*, and *recA*). Regarding the genomes from the SFV (São Francisco Valley), two sets of alleles repeat twice in each isolate, suggesting they can be clonal: B79 and B80; B44 and B52 (Appendix A).

The correlation index of tetranucleotides (TETRAs) clusters together MM_B02 with *A. dhakensis* and *A. hydrophila* (TETRA = 1.000 and 0.999, respectively) (Appendix A). All the other SFV (São Francisco Valley) isolates constitute a clear cluster (TETRA ≧ 0.994) encompassing *A. veronii* and other species (*A. fluvialis*, *A. lacus*, *A. jandaei*, *A. australiensis*, *A. allosaccharophila,* and *A. finlandensis*) (Appendix A). Despite the fact that TETRA analysis seems to not provide a sufficient resolution for unequivocally separating *A. veronii* from other species (such as *A. allosaccharophila* and *A. filandensis*), it can be seen that, considering the threshold of 0.999, the genomes WP2-S18-CRE-03, WP3-W19-ESBL-03, WP8-S18- ESBL-11, WP9-W18-ESBL-04, FC951, HD6448, and MM_B02 (SFV) could be suggested to not belong to the *A. veronii* species when compared with the reference FDAARGOS-632.

To overcome this, another classification method was performed to estimate the average nucleotide identity (ANI) between paired genomes in four ways: ANIb (BLAST), ANIm (MUMMER), OrthoANIb (BLAST), and OrthoANIu (USEARCH). Considering the threshold of 95%, 33/38 previously classified *A. veronii* genomes could be suggested to belong to the *A. veronii* species, and the genomes WP2-S18-CRE-03, WP3-W19-ESBL-03, WP8-S18-ESBL-11, WP9-W18-ESBL-04, and MM_B02 can be suggested to belong to another species. It is interesting to note that using this threshold (95%), all four ANI approaches used are completely concordant (Appendix A). Considering the threshold of 96%, 31/38 previously classified *A. veronii* genomes could be suggested to belong to *A. veronii* species. In addition to the aforementioned genomes, 1708-29120 did not reach the 96% threshold in all four ANI analysis, and 71506 genome did not reached the 96% threshold in the OrthoANIU analysis.

The local genome MM_B02 clusters with *A. dhakensis* KOR1, exhibiting average identity of nucleotides over 0.95 (Figure 1). The other local isolates constitute a cluster which includes the reference *A. veronii* genome FDAARGOS-632 (ANI > 0.95), and other deposited genomes of the species (Figure 1).

WP3-W19-ESBL-03, WP8-S18-ESBL-11, and WP9-W18-ESBL-04 genomes constitute a cluster with *A. allosaccharophila* FDAARGOS-933 (ANI > 0.95). The WP2-S18-CRE-03 genome constitutes a cluster with the *A. jandaei* Riv2 genome (ANI > 0.96) and *A. lacus* AE122 genome (ANI > 0.95), being related with the first one considering the ANI value. The other isolates used in this study (*n* = 25) constitute a cluster with the *A. veroni* reference genome FDAARGOS-632 (ANI > 0.95) (Figure 1). In this way, it can be suggested that the isolates WP3-W19-ESBL-03, WP9-W18-ESBL-04, WP8-S18-ESBL-11, and WP2-S18-CRE-03 do not belong to the *A. veronii* species, suggesting that they are misclassified in the NCBI database. With *A. veronii*, 71506 is the most divergent isolate from the others and the only isolate that presents ANI values lower than 96% (0.96) with the four methods being applied.

The isolates B79 and B80 are suggested to be clonal (ANI = 100%), as are the isolates B44 and B52. With the former, the samples were collected in different city points. With the latter, the samples were isolated from different species of hosts in the same location, with B44 isolated from *Pseudoplatystoma corruscans* and B52 from *Phractocephalus hemioliopterus*.

A maximum likelihood tree from 841 genes of the core genome was built (Figure 2). It can be observed that the genome MM_B02 from SFV, as with the genomes from NCBI WP2-S18-CRE-03, WP3-W19-ESBL-03, WP8-S18-ESBL-11, and WP9-W18-ESBL-04, is more related to the genomes of *A. dhakensis*, *A. jandaei*, and *A. allosaccharophila*. Inside the branch that includes only the *A. veronii* genomes, the 71506 isolate is the one that diverges most from the others. From this perspective, the results from phylogenic study are in accord with the ANI results that use 95% as the threshold for species classification. It is important to note that isolates from SFV, different from what was expected, do not constitute an only phylogenetic subgroup, and some of them are even closely related with isolates from China, as, for example, B42 and ZfB1; B48 and 183026; and B45 and GD21SC2322TT (Appendix A). The same can be noted about the isolates from other countries (Figure 2 and Appendix A).

### 2.2. Virulence

A total of 256 virulence genes were identified, and from these, 239 were present in the local VSF isolates. The number of genes common to all analyzed genomes was 236 (Appendix A), with seventeen exclusive to the complete *A. veronii* genomes deposited and three exclusive to local SFV genomes, including *hlyB*, which encodes Hemolysin B.

A total of 190 genes were shared among all isolates regardless of their source of isolation, whether from humans, fish, the environment, or other (Appendix A). In the fish group there were ten exclusive genes, while in the human group there were four exclusive genes. The other groups did not present exclusive genes. Fish isolates are of a bigger quantity in this study and present more exclusive genes than human isolates, including the ones which are involved with secretion systems and exotoxins.

A total of 224 of the 256 virulence genes identified involve functions of adherence, effector delivery systems, and motility, accounting for 59, 78 and 87 genes, respectively (Figure 3). Among the structures behind these functions, those that are present in all isolates are: type IV pili (*Tap*), polar flagellum, and the type II secretion system (*T2SS*) (Appendix A). In addition to the elements common to all isolates, present in the genomes from SFV and in some from NCBI are the lateral flagellum, type IV pili (*Flp* and *Msha*), type I pili, and type III and type VI secretion systems (*T3SS* and *T6SS*). Other genes are related to immune modulation, stress survival, metabolic and nutritional factors, exotoxins and exoenzymes, regulation, biofilm formation, and competitive advantage, among others.

A PCoA (Principal Coordinate Analysis) based on the presence and absence of virulence genes and their matrix reveals that the difference among the genomes is not explained by the type of host or local from which the specimen was collected (Appendix A), with local SFV isolates showing a similar virulence content to isolates from China (Appendix A). Interestingly, the single isolate from an asymptomatic host was collected from *Hirudo verdana* (leech) and presents a similar virulence content to those isolated from diseased fish (Appendix A).

Plasmids were identified on B44, B52, and B45 local isolates (Appendix A). One non-mobilizable plasmid was identified in the three isolates, whereas a conjugative plasmid of 144,051 bp was identified only in isolate B45. In this plasmid, three genes related to the type VI secretion system were identified: *vgrG1*, *AHA_RS09305*, and *hcp*. No genes related to virulence were identified in the 26 plasmids belonging to complete *A. veronii* genomes downloaded from the NCBI.

### 2.3. Resistance

There were 60 different genes conferring resistance to 22 classes of antimicrobials in the 33 *A. veronii* isolates analyzed. This resistance takes place through five mechanisms (Appendix A): antibiotic efflux (15 genes); antibiotic inactivation (27 genes); antibiotic target alteration (10 genes); antibiotic target replacement (6 genes); and antibiotic target protection (2 genes).

All isolates have genes associated with resistance to drugs from the following classes: Peptide, Carbapenem, Phenicol, Fluoroquinolone, Macrolide, Cephalosporin, Diaminopyrimidine, Penam, Penem, Monobactam, Sulfonamide, Cephamycin, Aminocoumarin, Nitroimidazole, and Tetracycline (Figure 4). Resistance to the last four classes is especially due to Efflux Pumps, which contain the largest number of genes in the genomes.

From all the antimicrobial resistance genes identified, 44 were exclusive to the genomes from NCBI, 1 exclusive to the local VSF genomes, and 15 genes were shared by the NCBI and local genomes (Appendix A). The isolates were assigned to four categories according to the isolation source: human, fish, the environment, and other. A total of 15 genes were shared among them. Fifteen genes were present only in humans, four only in fish, four in the environment, and four in others (Appendix A).

All analyzed *A. veronii* genomes harbor genes from the ABC transporters; RND efflux pump; Undercaprinlyl-Pyrophosphate-related proteins; *CphA*, *OXA*, and *Dfr*; and *MCR* (*mcr.3* and *mcr.7.1*) (Appendix A).

Of the plasmids obtained with the genomes of *A. veronii* from the NCBI, four were conjugatives, 11 were mobilizable, and 11 were non-mobilizable (Appendix A). In total, 29 genes in all were identified in 11 plasmids, with each conferring resistance to 13 classes of antimicrobials. They were grouped into four resistance mechanisms (Appendix A): antibiotic inactivation (27 genes in total), antibiotic target replacement (11 genes in total), antibiotic efflux (4 genes in total), and antibiotic target protection (4 genes in total). Among them, it is worth highlighting three conjugative plasmids: pASP-a58 from the lineage AVNIH1; p158496 of the SW3814 lineage, isolated from an environment in the United States; and pWP8-W19-CRE-03_1, from the WP8-W19-CRE-03 lineage, isolated from an environment in Japan. The first encodes 13 different genes conferring resistance to 10 classes of antimicrobials, the second encodes 8 different genes conferring resistance to 7 classes, and the last one has 5 genes that confer resistance to 5 classes. No gene related to antimicrobial resistance was identified in plasmids from the SFV.

## 3. Discussion

Nine genomes of SFV isolates were submitted to different tools for taxonomic classification. The tetranucleotide signature (TETRA) revealed that the SFV genome, MM_B02, along with six other NCBI genomes, may not be *A. veronii* when compared to the reference genome FDAARGOS-632. However, confusing results can be observed with this method, as some genomes are related to more than one species. As pointed out by Richter and Rosselló-Móra [21], TETRA shows irregularities when applied to fragmented genomes, being more inaccurate than other methods, such as ANI, for example.

The average nucleotide identity (ANI) showed greater resolution in genome separation, including for reference sequences from other species (Figure 1). The results of this analysis show that eight isolates from SFV were *A. veronii*, while one isolate (MM_B02, from human) was misidentified and belongs to the *Aeromonas dhakensis* species. What stands out is the fact that there is, also, amongst the whole genome data obtained from the NCBI, misidentified genomes were deposited, and they were WP2-S18-CRE-03, WP3-W19-ESBL-03, WP8-S18-ESBL-11, and WP9-W18-ESBL-04. To the best of our knowledge, only the first genome has been explicitly highlighted as erroneously identified in a previously published study [6].

Identification errors in *Aeromonas* species have already been pointed out in other studies [17,18]. Correct identification of pathogens is critical to understanding their epidemiology and their true impact on human, animal, and environmental health [17]. In this sense, the tools applied to calculate the ANI from genomes, whether complete or drafts, proved to be useful and consistent with the 95% threshold [17] for *A. veronii* (Appendix A), although 96% is suggested to be more suitable for other species of the genus *Aeromonas* [22,23].

Interestingly, clonal isolates among *A. veronii* from the SFV occurred twice (Figure 1), which is noteworthy because they were sampled at different points in the region, as was the case for B79 and B80, and for hosts of different species, as for B44 and B52. This situation reflects what can be observed when analyzing the general phylogeny of the genomes used in this study (Figure 2 and S3), as it is possible to notice that isolates from different countries, or different host species, may be more closely related to each other, for example, isolate B48 and 183026. A similar situation was highlighted by Liu and his collaborators [6], a study which revealed the ease with which *A. veronii* can be spread locally and globally, in addition to its ability to infect different types of hosts.

Most of the virulence genes are common to SFV and NCBI isolates (Figure 3), and they were mainly involved with adherence, secretion systems, and motility. Most of the genes involved with adherence in this species encode structures such as type IV pili (*Tap*, *Msha*, and *Flp*), which are essential during the host colonization step, especially *Msha* type IV pili, absent only in isolate FC951, which is necessary for *A. veronii* adherence and biofilm formation by the species [24]. Interestingly, genes encoding type I pili were also identified in nine isolates, including five from the SFV. This filamentous adhesin is one of the main virulence factors in uropathogenic *Escherichia coli* and is responsible for mediating attachment to a variety of host surfaces and tissues [25]. In *Aeromonas salmonicida*, type I pili contributes to the colonization of fish [26]. Most isolates containing this structure are from fish; one of each of the others are from humans, alligators, and the environment.

Most *Aeromonas* species, including *A. veronii*, can move via a single polar flagellum [1], thus, genes that encode this component are conserved in the species genomes, according to the results of the analyses. The lateral flagellum, on the other hand, is found with an approximate frequency of 50–60% in mesophilic species associated with diarrheal diseases [27]. Among the whole genomes from the NCBI used in this study, only 44% have a set with more than 30 genes encoding this structure, while all isolates from the SFV have such a set of genes. In addition to motility, the importance of this component for pathogenicity is also due to its participation in the stages of adhesion and colonization of the host’s intestine, invasion, movements on surfaces, and biofilm formation [27,28].

Bacteria use secretion systems to secrete proteins that enhance adherence to host cells, obtain resources from environmental niches, and even directly intoxicate target cells [29]. *Aeromonas* spp. utilize *T1SS*, *T2SS*, *T3SS*, and *T6SS* to transport virulence factors, toxins, and effectors during host fish infection [30]. In this work, genes coding for type II, type III, and type VI secretion systems were identified. *T2SS* is conserved in *A. veronii* [8,29] and is used to move substrates across the bacterial cell membrane, releasing them directly into the environment, with the substrates then participating in nutrient acquisition, biofilm formation, and overall pathogenicity [31]. The secreted products of this system include proteases, lipases, phosphatases, and toxins, among others [32]. Genes encoding aerolysin (*aerA*) and metalloprotease (*stcE*), both products secreted by *T2SS* [32], were found in all isolates from the SFV and almost all isolates from the NCBI. Aerolysin is a pore-forming toxin [33] and acts by disrupting the fish intestinal barrier, being a key virulence factor in *A. veronii* [4]. StcE metalloproteases are secreted by Enterohaemorragic-*Escherichia coli* (EHEC) and are related to reduced mucus levels, facilitating adherence to human hosts [34].

Differently from what happens in *T2SS*, *T3SS* and *T6SS* can both deliver effectors directly into target cells [29]. Gram-negative plants and animal pathogens use T3SS to inject virulence factors directly into the cells of their hosts [35]. *T3SS* is expressed in important human enteric pathogens, which translocate effectors to the interior of their hosts, which act on the cytoskeleton and cell traffic, in addition to important cellular pathways, aiding in the establishment of infections and blocking phagocytosis [36]. The effectors of this system commonly found in *A. veronii* are the AexT and AexU toxins [37]; however, the *T3SS* effectors identified in the present study were phosphatases encoded by the *ati2* and *aopH* genes, with neither of them found in genomes from the SFV, despite the presence of T3SS in these isolates. Genes encoding T6SS are present in six out of eight genomes from the SFV. This secretion system exports effectors which act in both bacterial and eukaryotic cells, leading to the death of competing and/or host cells [38]. Other functions attributed to T6SS involve adaptation to stress conditions caused by temperature or pH changes, as well as the uptake of metal ions in response to oxidative stress [38,39].

Other virulence elements identified in *A. veronii* genomes include LPS (lipopolysaccharides), LOS (lipooligosaccharides), capsule, genes associated with adherence (*Hsp60*, *IlpA*, *EF-Tu*), genes related to iron uptake (*entB*, *entE* and *Fur*), and *luxS*. The *luxS* gene, absent only in the isolate B45 from the SFV, encodes an S-ribosilhomocysteinase that plays a key role in the synthesis of the AI-2 autoinducer [40]. This autoinducer plays an important role in interspecies communication, especially quorum sensing, thus influencing biofilm formation, virulence, stress resistance in different environments, and antibiotic resistance [40,41,42,43].

Genes encoding resistance to 22 classes of antibiotics were identified in the *A. veronii* genomes used in this study. Part of this resistance can be attributed to the presence of gene coding for beta-lactamases, since many *Aeromonas* species are naturally resistant to some beta-lactam antibiotics [1]. It is important to draw attention to the Extended Spectrum Beta-lactamases (ESBL), identified only in isolates GD21SC2322TT and 1708-29120, from China, which are responsible for the resistance of some pathogens to a wide spectrum of beta-lactam antibiotics, except for carbapenems [44]. However, this class of antibiotics can be inactivated by Metallo-Beta-Lactamases (MBLs) [45], such as those encoded by the *CphA* gene family, which is common to all *A. veronii* genomes investigated in this study (Appendix A).

Even more important is the colistin resistance, which is considered one of the last resources to treat infections caused by multidrug-resistant pathogens [13]. The *mcr-3* and *mcr*-7 genes, which encode phosphoethanolamine transferase, were identified in this work, the last one being present in all isolates. The *mcr*-3 gene, present only in isolates from the NCBI, was characterized by Yin et al. in 2017, who pointed out its widespread prevalence among *Enterobacteriaceae* and *Aeromonas* [13]. The *mcr*-7 gene was first identified in *Klebisiella pneumoniae* from chickens and is closely related to *mcr*-3 from *Aeromonas*, suggesting a likely origin in this genus [14,15]. Apparently, *Aeromonas* spp. may be likely reservoirs for *mcr* genes, disseminating them through horizontal gene transfer in the most diverse aquatic environments in which they are present [13,15].

Several genes identified in the genomes analyzed can confer resistance to antibiotics such as Phenicol, Fluoroquinolone, Macrolide, Tetracycline, Diaminopyrimidine, Sulfonamide, Aminocoumarin, and Peptides, which is due, especially, to the fact that some of them code for antibiotic efflux systems. The ability to rapidly export drugs out of bacterial cells is an important contributing factor to antimicrobial resistance in pathogens [46]. Three families of efflux pumps can be identified in the isolates of this work, namely: MFS, RND, and ABC. The last two are present in all *A. veronii* genomes and may contribute to multidrug resistance in the species when they are overexpressed, especially the RND efflux pump, and they are considered the most clinically significant [46,47]. MFS transporters, although considered ubiquitous in all domains of life [46], did not have genes identified in all the genomes of this study, being present in only three of the eight isolates from the SFV, and they were B48, B79, and B80. This family tends to be a specific substrate [47], highlighting that genes *tet(A)* and *tet(E)*, coding for the efflux of tetracyclines, and *floR* were involved with the efflux of Fenicol, both of which can be transferred horizontally through mobile genetic elements [48,49].

These efflux pumps contribute to prospective resistance, according to CARD information [50], to 18 classes of drugs. The mere presence of efflux pumps may not be enough to confer resistance, since resistance depends on other factors, such as expression [46,47]. However, resistance to some classes of antibiotics may occur due to more than one resistance mechanism encoded in the same genome, as occurs with beta-lactams in isolate GD21SC2322TT, for example.

The spread of antibiotic resistance involves the horizontal transfer of genes through mobile genetic elements, especially plasmids, via conjugation [51]. No genes related to antibiotic resistance were identified in plasmids isolates from the SFV; however, it is necessary to highlight some plasmids deposited with complete genomes from the NCBI. An example of this is the conjugative plasmid pASP-a58, from the environmental isolate AVNIH1, which encodes thirteen genes related to four resistance mechanisms (antibiotic inactivation, target replacement, efflux pump, and target protection) and confers resistance to ten antibiotics classes, including Carbapenem, Phenicol, and Aminoglycoside. Multiple resistance genes on a plasmid such as this help to spread multi-drug resistance more easily [51], especially in the case of conjugative plasmids, since they contain enough elements to carry out their own transfer [52].

Differently from Tekedar et al. [8] and Liu et al. [6], in the present study, *T1SS*, *T4SS*, and *T5SS* were not identified, as was the case in the study by Prediger et al. [7], although in this case a single isolate was characterized. On the other hand, type I pili was only identified in the present study and in the one by Prediger et al. [7]. This difference in the virulence elements found can be caused by several factors, such as tools and methods performed during annotation, database content, coverage percentage, and established identity, among others. This reveals the need to create tools and databases that are personalized with reference genes for some taxa that have gained importance as emerging pathogens, such as *Aeromonas* [2,4,5,6,16].

The *A. veronii* species, like other species in the genus, can adapt very well to the selective pressures of antibiotics use, since its pan-genome is open and exhibits high genomic plasticity, being influenced by mobile genetic elements through the horizontal transfer of genes [8,10,16,53]. This genome plasticity also influences, among other aspects, the virulence of *Aeromonas* spp. [53,54].

*Aeromonas veronii* is known to infect fish, cause mortality, and negatively impact aquaculture [2,3,4]; however, more attention should be given to the species, as it has increasingly emerged as a pathogen associated with cases of intestinal diseases in humans [5,6,55]. According to the analyses of the present study, it is possible to verify that fish isolates, such as B48, MS-18-37, and X11, are closely related to human isolates (Appendix A). Furthermore, the virulence gene content of isolates from different isolation sources, including diseased humans, are similar (Appendix A). Given its characteristics [1,2], the species has the potential to infect humans from food sources, especially fish [56], and from water used for consumption [57]. *A. veronii* is a mesophilic species of the genus *Aeromonas* [2] able to proliferate at higher temperatures, and some studies, such as the one by Yuwono et al. [5], reported a higher frequency of *Aeromonas* infections in warmer seasons.

In addition to its pathogenic potential, *A. veronii* may play an important role as a source of antibiotic resistance, maintaining and disseminating antimicrobial resistance genes in the aquatic environment, including *mcr* genes [10,15,51,58,59]. It is important to note that fish and human isolates practically share resistance to the same classes of antibiotics (Figure 4), and human isolates show an even greater amount of resistance genes, even though they are fewer in number (Appendix A). When thinking about surveillance systems for antimicrobial resistance, which is considered one of the main threats to public health, it is necessary to include data from human and animal pathogens, as well as environmental isolates, following a One Health approach [20]. In this sense, it is also necessary to include emerging species as potential threats to public health; these species already play an important role in the infection of animals or are disseminated in the environment, as with *A. veronii*.

Finally, isolates from fish native to the São Francisco Valley (SFV) showed a similar virulence profile to fish isolates from other parts of the world, mainly those from China. The antimicrobial resistance content was also similar, although lower, to the content identified for complete genomes of *A. veronii* deposited at the NCBI, indicating that antimicrobial resistance genes, or resistant isolates, are now widely disseminated throughout the world. As for already deposited genomes, there was also an identification error, which was corrected with genomic sequencing and the application of tools that classify by comparing the entire genome, especially the ANI method, revealing flaws in phenotypic classification systems. In general, the gene content related to virulence and resistance to antibiotics, as well as the ability to infect hosts other than fish, makes *A. veronii* a species with zoonotic potential that should be on the radar of surveillance systems, especially those based on the One Health initiative.

## 4. Materials and Methods

### 4.1. Genomes from the São Francisco Valley (SFV)

Nine isolates recovered from fish, humans, and the environment (Table 1) were classified as *Aeromonas veronii* based on the automated PhoenixTM system (Becton Dickinson, Franklin Lakes, USA). The fish and environmental isolates were collected from 2017–2019, and the human isolate was collected in 2021. All isolates were maintained in a BHI broth added with glycerol in −80C until analysis. DNA was isolated using a salting-out protocol [60]. In summary, 500 µL of lysis solution (10 mM Tris-HCl pH 8; 10 mM EDTA pH 8; 100 mM NaCl; 0.5% SDS; and 2 µg of proteinase K) was added, followed by vortexing and incubation in a dry bath for 4–6 h at 55 °C. Then, 210 µL of TE (pH 7.6) and 240 µL of NaCl 5M were added, with ice incubation for 10 min. Samples were centrifuged for 15 min at (16,000× *g*), and 750 µL of supernatant was collected and added to new microtubes. In total, 750 µL of isopropyl alcohol (100%) was added, and samples were centrifuged (15min; 16,000× *g*) and the supernatant then discarded. In total, 500 µL of ethanol (70%) was added to the samples, followed by incubation (5 min) and centrifugation (5 min; 16,000× *g*). The supernatant was discarded, and 100 µL of H_2_O + RNAse (10 µg per mL) was added to the samples, which was subsequently incubated at 37 °C for 1 h.

DNA obtained was submitted to sequencing using NEBNext1 Fast DNA Fragmentation and Library Preparation Kit (New England Biolabs Inc., Ipswich, MA, USA) following the manufacturer’s protocol. Whole-genome sequencing was performed using an Illumina HiSeq 2500 platform (Illumina Inc., San Diego, CA, USA) with coverage above 100×.

### 4.2. NCBI Genomes

All available complete *A. veronii* genomes (*n* = 29) were downloaded along with their plasmids from the NCBI (Appendix A). Additionally, thirty reference genomes from other *Aeromonas* species were used for taxonomy, and six randomly selected complete genomes (three *A. dhakensis* and three *A. jandaei*) were used for phylogeny analysis (Appendix A).

### 4.3. São Francisco Valley Genome Assembly

Raw data were submitted to an assembly pipeline that combines different approaches (Available online: https://github.com/engbiopct/assembly-hiseq, accessed on 23 April 2022). In summary, FastQC [61] was used in to assess data quality, AdapterRemoval v.2 [62] to remove the adapters, and KmerStream [63] to estimate better k-mer values within the proposed size (7–127, two-by-two). The genome assembly was performed with SPAdes [64], Edena [65], and Unicycler [66], and CD-HIT [67] was used to reduce computational costs and facilitate the assembly of sequences. The best assembly was chosen considering the metrics assessed using QUAST (Appendix A) [68].

### 4.4. Plasmids

Mob-suite [69] was used to identify potential plasmids in the assembled genomes and separate them from chromosome sequences, with the MOB-recon tool used to reconstruct and perform typing of genomes sequenced in this study, and MOB-typer used to perform sequence typing for plasmids downloaded from the NCBI. The Plasmidfinder tool from Abricate version 1.0.1 (Available online: https://github.com/tseemann/abricate, accessed on 12 July 2022) was used to search for replicons.

### 4.5. Annotation

Prokka version 1.14.6 [70] was used with the genetic code 11 (—gcode 11) and default values for the other under-standard parameters to perform the annotation of possible features and CDSs for genomes.

### 4.6. Typing and Genomes Classification

The mlst tool (Available online: https://github.com/tseemann/mlst, accessed on 16 May 2022) was used to perform genome typing using multi-locus sequence typing (MLST) following the pubMLST scheme (Available online: https://pubmlst.org/bigsdb?db=pubmlst_aeromonas_seqdef, accessed on 16 May 2022). To calculate the average nucleotide identity (ANI) among the genomes, the pyANI and [71] (Available online: https://github.com/widdowquinn/pyani, accessed on 21 May 2022) and OrthoANI tools were utilized [72] (Available online: https://www.ezbiocloud.net/tools/orthoani, accessed on 2 June 2022 and Available online: https://www.ezbiocloud.net/tools/orthoaniu, accessed on 6 June 2022). The first one, pyANI version 0.2.11, estimated ANI using BLAST (ANIb) and Mummer (ANIm) in the scheme all vs. all. OrthoANIu version 1.2 was used to estimate ANI applying Usearch with the scheme all vs. all, meanwhile OrthoANIb version 1.40 was applied using BLAST with the scheme all vs. all. The reference species was *A. veronii* genome FDAARGOS-632 (Available online: https://www.ncbi.nlm.nih.gov/assembly/GCF_008693705.1/, accessed on 2 May 2022). The pyANI tool was used to calculate the correlation indexes of tetranucleotides signatures (TETRAs) among the genomes. Genomes were considered clonal in this study if one of the following criteria was met: (1) Present the same alleles for all genes (*gyrB, groL, gltA, metG, ppsA,* and *recA*); (2) ANIm = 1; (3) ANIb = 1; (4) OrthoANIb = 100%; (5) OrthoANIu = 100%; (6) TETRA = 1.

### 4.7. Phylogeny

Prokka output files were submitted to Roary v. 3.13.0 [73], and the core genome was aligned with MAFFT v. 7.505 [74]. IQ-TREE v. 2.2.03 [75] was performed to build the phylogenetic tree from the core genome with the substitution model GTR + F+I + G4 and 1000 replicates from Bootstrap, and iTOL v. 6 (Available online: https://itol.embl.de/, accessed on 15 August 2022) was used to edit and annotate the tree.

### 4.8. Virulence and Resistance Genes Analysis

Abricate version 1.0.1 (Available online: https://github.com/tseemann/abricate, accessed on 12 July 2022) was used to identify virulence and resistance genes in genomes and plasmids, with a minimum of 60% coverage and identity. The databases used for gene identification were VFDB [76] and CARD [50]. A presence/absence matrix was constructed and used to generate a dissimilarity matrix using the Jaccard index [77,78] and these data were used to perform PCoA (Principal Coordinate Analysis) [79] using the cmdscale function from R (Available online: https://cran.r-project.org/, accessed on 3 January 2022).

## 5. Conclusions

This study suggests that ANI is the method with the highest resolution for separating species using the genome information once phenotypic methods are not completely reliable for classifying *Aeromonas* species. Additionally, it can be seen that a widespread distribution of virulence genes identified in isolates originated from different sources and geographic origins. Additionally, the antibiotic resistance profile revealed that *A. veronii* harbors genes related to several classes of antimicrobial resistance genes, including those that confer resistance to antimicrobials used as the last line of defense against resistant bacteria such as carbapenem and colistin. Taken together, our results pinpoint that *A. veronii* is an emerging pathogen that deserves more attention from surveillance systems from a One Health perspective, since it can be involved in local and worldwide dispersion of virulence and resistance, thus causing fatal diseases in humans and fishes.

## Figures and Tables

**Figure 1 antibiotics-12-01039-f001:**
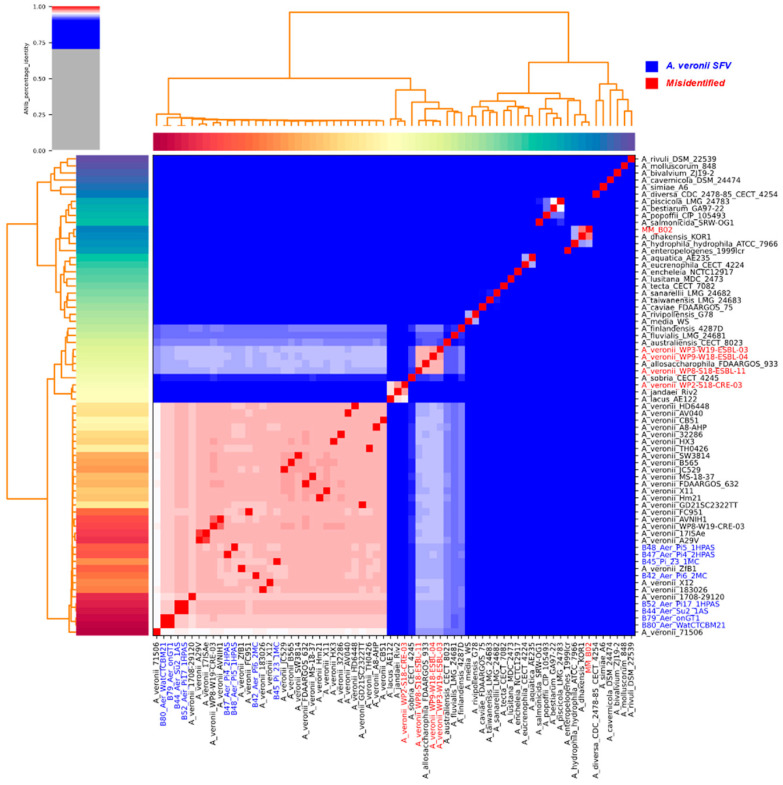
Heatmap illustrating the genomes relationship according to the estimated pairwise nucleotide average identity. Warmer colors represent values closer to 1. *A. veronii* isolates from SFV are highlighted in blue; misidentified isolates are highlighted in red.

**Figure 2 antibiotics-12-01039-f002:**
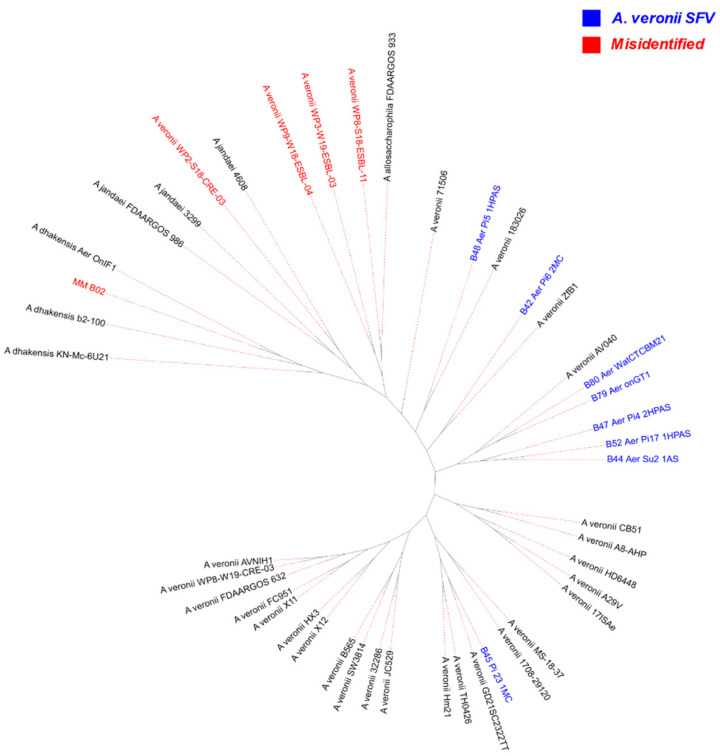
Tree built from core genome SNPs (841 genes). Misidentified isolates are highlighted in red; *A. veronii* isolates from SFV are highlighted in blue.

**Figure 3 antibiotics-12-01039-f003:**
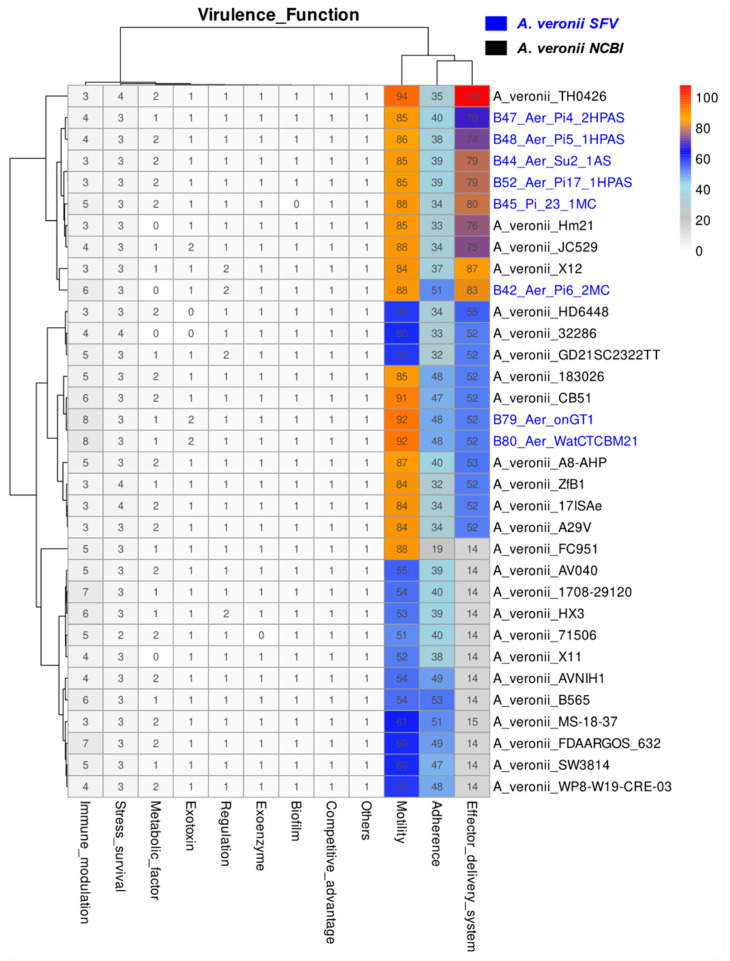
Gene numbers belonging to each category of function related to the virulence in each genome. *A.veronii* isolates from SFV highlighted in blue.

**Figure 4 antibiotics-12-01039-f004:**
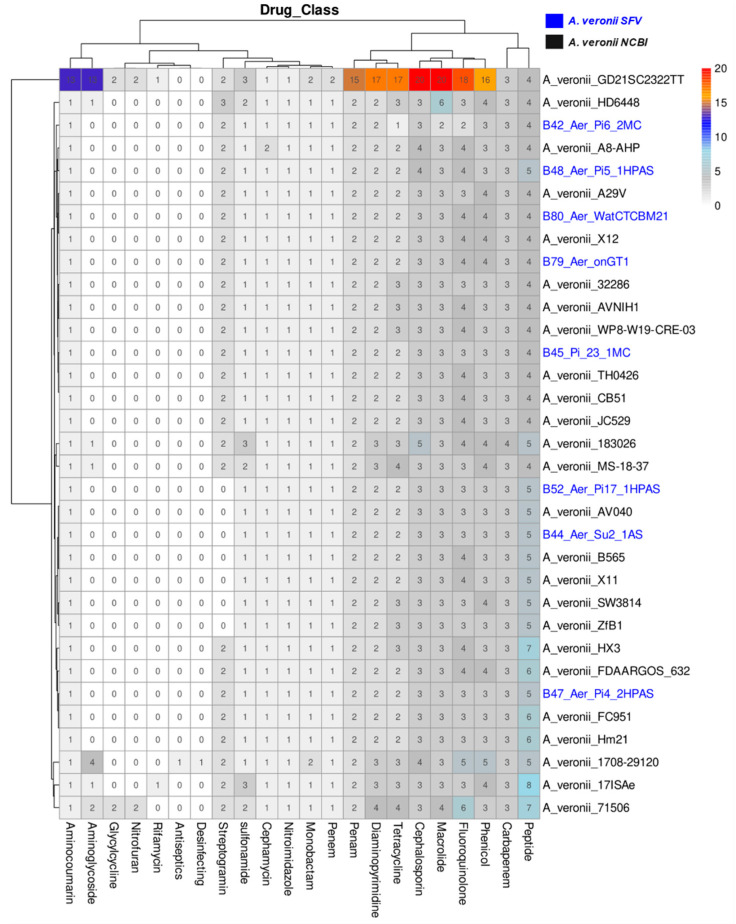
Genes number conferring drug class resistance in each isolate. Isolates from SFV are highlighted in blue.

**Table 1 antibiotics-12-01039-t001:** Information related to the *Aeromonas veronii* isolates from fishes, humans, and the environment in the São Francisco Valley.

ID	Isolates	Isolates	Host	Location	Patient	Plasmids
B42	PRJNA590960	B42_Aer_Pi6_2MC	*Phractocephalus* *hemioliopterus*	CODEVASF *	kidney	-
B44	PRJNA594379	B44_Aer_Su2_1AS	*Pseudoplatystoma corruscans*	CODEVASF *	kidney	Yes
B45	PRJNA590964	B45_Pi_23_1MC	*Phractocephalus* *hemioliopterus*	CODEVASF *	kidney	Yes
B47	PRJNA590505	B47_Aer_Pi4_2HPAS	*Phractocephalus* *hemioliopterus*	CODEVASF *	kidney	-
B48	PRJNA594423	B48_Aer_Pi5_1HPAS	*Phractocephalus* *hemioliopterus*	CODEVASF *	kidney	-
B52	PRJNA595563	B52_Aer_Pi17_	*Phractocephalus* *hemioliopterus*	CODEVASF *	kidney	Yes
		1HPAS				
B79	PRJNA590958	B79_Aer_onGT1	*Oreochromis* *niloticus*	Instituto Federal	kidney	-
B80	PRJNA594352	B80_Aer_	Environmental (water)	CODEVASF *	Not applicable	-
		WatCTCBM21				
MM_B02	-	MM_B02	*Homo sapiens*	Teaching Hospital	Soft tissues infection	-

* Development Company of the São Francisco and Parnaiba Valleys.

## Data Availability

Genome sequence data are available at the National Center for Biotechnology Information (NCBI).

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
