# Peer review of "Genomic Characterization of *Aeromonas veronii* Provides Insights into Taxonomic Assignment and Reveals Widespread Virulence and Resistance Genes throughout the World"

_antibiotics, 2023, doi:10.3390/antibiotics12061039_

Round 1
Reviewer 1 Report
This is an article about Aeromonas veronii, a Gram-negative bacterial species that causes disease in fish and has been described in human infections. As a general comment, I think your article is relevant. However, you need to be more transparent about the methods. In this journal, your audience is microbiologists, that are not aware of all analysis methods. Therefore, you need to justify, explain, and provide accurate references for your methodology, and the different index that you use.
You will find below some specific comments :
Affiliation :
check affiliation, some informations are missing.
Materiel and methods
Do you have data of sensitivity and specificity of phoenix TM system to identify Aeromonas veronii.
Did you take all the A. veronii available in NCBI (if not please justify)? How did you select the 36 additional genomes from other species?
Could you justify the use of your assembly pipeline (assembly Hiseq). What is the purpose of using three genome assembly software? Does it improve the N50? Is there any published comparison between methods (using just spade or just unicycler of just Edena vs the three of them?)
You should also add the quality data of your sequence (N50, coverage…)
Citation of pyANI ans orthoANI are not correct, please clarify and add the published reference with the github link to get the code.
For pyANI you need to reference ritchard et al. (2016) "Genomics and taxonomy in diagnostics for food security: soft-rotting enterobacterial plant pathogens" Anal. Methods 8, 12-24 DOI: 10.1039/C5AY02550H
Line 469 : Which GFF files are you talking about?
Results :
Line 95 : “suggesting they can be clonal” Can you define what is clonal in the context of your study (this should be part of your material and method).
Can you also provide information on the correlation index of TETRA.
The name of the genes are not written correctly, it should be in italic.
The information on the plasmid are sparse. Could you describe what analysis you performed with MobSuite. Did you find replicons (with plasmid finder?). Were you able to circularize some plasmid. Could you also compare the plasmids to some find in NCBI (in other species)?
Do you have data on the resistance phenotype? (antimicrobial susceptibility ?)
Comments on tables and figure :
Table 1 : Is it possible to reduce the size of the police to make the table easier to read
What is CODEVASF?
Your text should be revised for English. I am not an English native speaker but I found some minor errors.
Author Response
|
Reviewer #1: |
|
|
Comment |
Alteration/Observation |
|
General Comment: This is an article about Aeromonas veronii, a Gram-negative bacterial species that causes disease in fish and has been described in human infections. As a general comment, I think your article is relevant. However, you need to be more transparent about the methods. In this journal, your audience is microbiologists, that are not aware of all analysis methods. Therefore, you need to justify, explain, and provide accurate references for your methodology, and the different index that you use. |
We appreciate your considerations and believe that they will contribute to improving our manuscript. The manuscript was completely revised, and alterations were made to be more transparent about the methods.
|
|
check affiliation, some information are missing. |
Zip code was added to the affiliation in the reviewed version. |
|
Do you have data of sensitivity and specificity of phoenix TM system to identify Aeromonas veronii? |
Despite of the fact that we don’t have original data about classification of Aeromonas veronii using the PhoenixTM System, a study aiming to compare six different automated classification systems showed an accuracy of 86.7% (26/30) in the classification of A. veronii isolates (Lamy et al., 2010). This study was included as reference in the text (L.77)
19. Lamy, B.; Laurent, F.; Verdier, I.; Decousser, J.W.; Lecaillon, E.; Marchandin, H.; Roger, F.; Tigaud, S.; de Montclos, H.; Kodjo, A. Accuracy of 6 Commercial Systems for Identifying Clinical Aeromonas Isolates. Diagn. Microbiol. Infect. Dis. 2010, 67, 9–14, doi:10.1016/j.diagmicrobio.2009.12.012. |
|
Did you take all the A. veronii available in NCBI (if not please justify)? How did you select the 36 additional genomes from other species? |
We used all complete A. veronii genomes (n=29) available in NCBI when the analysis was performed. To perform taxonomy, 30 reference genomes (the reference genome defined by NCBI) from other Aeromonas species were included and for phylogeny, we also used six (three A. dhakensis and three A. jandaei) randomly selected complete genomes available in NCBI. The text was rewritten (L.460-465) in the manuscript to better describe the genomes used in this study. |
|
Could you justify the use of your assembly pipeline (assembly Hiseq). What is the purpose of using three genome assembly software? Does it improve the N50? Is there any published comparison between methods (using just spade or just unicycler of just Edena vs the three of them?) |
The pipeline used doesn’t combine results from the assemblers. Genomes were assembled independently with each assembler and the best assembly (considering the metrics assessed using QUAST) was chosen. To avoid misunderstanding, the text was rewritten (L.469-477) in the manuscript. |
|
You should also add the quality data of your sequence (N50, coverage…) |
A new supplementary table (Table S7) was included to incorporate sequenced genomes quality data. Reference for the Table S7 was added in the main manuscript (L.476-477). |
|
Citation of pyANI ans orthoANI are not correct, please clarify and add the published reference with the github link to get the code. |
We included the following references: pyANI: 71. Pritchard, L.; Glover, R.H.; Humphris, S.; Elphinstone, J.G.; Toth, I.K. Genomics and Taxonomy in Diagnostics for Food Security: Soft-Rotting Enterobacterial Plant Pathogens. Anal. Methods 2016, 8, 12–24, doi:10.1039/c5ay02550h. OrthoANI: 72. Lee, I.; Kim, Y.O.; Park, S.C.; Chun, J. OrthoANI: An Improved Algorithm and Software for Calculating Average Nucleotide Identity. Int. J. Syst. Evol. Microbiol. 2016, 66, 1100–1103, doi:10.1099/ijsem.0.000760.
We also included the repository links for the tools: pyANI: https://github.com/widdowquinn/pyani OrthoANI: https://www.ezbiocloud.net/tools/orthoani https://www.ezbiocloud.net/tools/orthoaniu
|
|
For pyANI you need to reference ritchard et al. (2016) "Genomics and taxonomy in diagnostics for food security: soft-rotting enterobacterial plant pathogens" Anal. Methods 8, 12-24 DOI: 10.1039/C5AY02550H |
The reference was included. |
|
Line 469: Which GFF files are you talking about? |
The GFF files referred in the first version of the manuscript are the Prokka output files that were used as input for Roary. The text was rewritten (L.508) in the manuscript.
|
|
Line 95 : “suggesting they can be clonal” Can you define what is clonal in the context of your study (this should be part of your material and method). |
Genomes were considered clonal in this study if one of the following criteria was met: (1) Present the same alleles for all genes (gyrB, groL, gltA, metG, ppsA, and recA); (2) ANIm = 1; (3) ANIb = 1; (4) OrthoANIb = 100%; (5) OrthoANIu = 100%; (6) TETRA = 1. This sentence was included in the material and methods (L.504-507). |
|
Can you also provide information on the correlation index of TETRA. |
The correlation indexes of tetra nucleotide signatures (TETRA) is an alignment-independent metric based on the assumption that every species is unique at the nucleotide level (Santana et al., 2019). In brief, according to Teeling et al., 2004, frequencies of all 256 possible tetranucleotides and their corresponding expected frequencies are computed for each genome and the differences between observed and expected values were transformed into z-scores for each tetranucleotide. The expectation is that two closely related genomes will show a similar distribution of the usage of these signatures (Richter et al., 2009). Teeling et al., 2004. Application of tetranucleotide frequencies for the assignment of genomic fragments. Environmental Microbiology, 6: 938-947. https://doi.org/10.1111/j.1462-2920.2004.00624.x Sant’Anna et al.,2019, Evelise Bach, Renan Z. Porto, Felipe Guella, Eduardo Hayashi Sant’Anna & Luciane M. P. Passaglia (2019) Genomic metrics made easy: what to do and where to go in the new era of bacterial taxonomy, Crit Rev Microbiol. 45(2):182-200. doi: 10.1080/1040841X.2019.1569587. Richter & Rosselló-Móra, 2019. Shifting the genomic gold standard for the prokaryotic species definition. PNAS, 106 (45) 19126-19131. https://doi.org/10.1073/pnas.0906412106
TETRA results were rewritten in the manuscript to provide more information related to this analysis (L.103-116). |
|
The name of the genes are not written correctly, it should be in italic. |
The manuscript was completely revised, and alterations were made to correct gene names. |
|
- The information on the plasmid are sparse. Could you describe what analysis you performed with MobSuite. - Did you find replicons (with plasmid finder?). - Were you able to circularize some plasmid. - Could you also compare the plasmids to some find in NCBI (in other species)? |
Mob-suite was used to identify potential plasmids in the assembled genomes and separate them from chromosome sequences with the MOB-recon tool used to reconstruct and perform typing of genomes sequenced in this study and MOB-typer used to perform sequence typing of plasmids downloaded from NCBI. Plasmid finder tool from Abricate version 1.0.1 (https://github.com/tseemann/abricate) was used to search for replicons. This information was included in the methods section (L.495-499) Non-mobilizable plasmids identified in isolates B44, B45 and B52 were circularized. The conjugative plasmid identified in isolate B45 was not circularized. This information was included in Table S5. Considering that our manuscript aims to describe virulence and resistance aspects of A. veronii genomes, we also focused our plasmid analysis in comparing the plasmids sequenced in our study with those associated to other A. veronii genomes and because of this, we didn’t perform interspecific comparisons. |
|
Do you have data on the resistance phenotype? (antimicrobial susceptibility ?) |
Considering that Aeromonas imposes a challenge to long term maintenance under laboratory conditions, we were not able to perform detailed and consistent phenotypic characterization of these isolates. |
|
Table 1 : Is it possible to reduce the size of the police to make the table easier to read |
Table 1 was altered. |
|
What is CODEVASF? |
Development Company of the San Francisco and Parnaiba Valleys (Companhia de Desenvolvimento dos Vales do São Francisco e do Parnaíba). The company operates in the economic development of hydrographic basins that extend across several states in Brazil. The description was included in Table 1. |
|
Your text should be revised for English. I am not an English native speaker but I found some minor errors. |
The manuscript was completely revised. |
Reviewer 2 Report
The main question addressed by the research: Genomic characterization of Aeromonas veronii provides insights into taxonomic assignment and reveals widespread virulence and resistance genes throughout the world. There is important, because the role of this species as a pathogen associated with antibiotic resistance capable of causing disease in humans is increasingly emphasized.
This topic «Genomic characterization of Aeromonas veronii provides insights into taxonomic assignment and reveals widespread virulence and resistance genes throughout the world » relevant in the field study of the molecular genetic characteristics of this microorganism, its virulence and antibiotic resistance.
This article summarized last information about 9 sequenced isolates from fish, environment, and human from São Francisco Valley (Brazil) isolates by comparing them with complete genomes deposited in NCBI.
The conclusion helps the reader evidence and arguments presented for understand the important point of Genomic characterization of Aeromonas veronii.
The references appropriate. The number of references (73) meet the requirements of the journal for articles, in addition, the number of sources five years ago (2018-2023) is 47,9% (35), which is enough.
The figures and tables are very informative and illustrative.
In this article presents strains isolated from hospitalized people, but there is no conclusion of the ethical committee.
In this article no information about statistical evidence of the data.
In the Table S5 presented information about Putative plasmids identified in São Francisco Valley isolates, but I did not find of those plasmids submitted into the Gen-Bank database.
In the Figure S11 and in the article no information about resistance mechanism in plasmids in strains Aeromonas veronii, isolated from São Francisco Valley (Brazil).
Author Response
|
Reviewer #2: |
|
|
Comment |
Alteration/Observation |
|
In this article presents strains isolated from hospitalized people, but there is no conclusion of the ethical committee. |
We appreciate your considerations and believe that they will contribute to improving our manuscript. Information related to ethical committee was included (L.595-599). |
|
In this article no information about statistical evidence of the data. |
We included more details to the methodology. |
|
In the Table S5 presented information about Putative plasmids identified in São Francisco Valley isolates, but I did not find of those plasmids submitted into the Gen-Bank database. |
Putative plasmids were deposited together with the genomes. For the analyses performed, we separate them from chromosome sequences with the MOB-recon tool from Mob-suite. |
|
In the Figure S11 and in the article no information about resistance mechanism in plasmids in strains Aeromonas veronii, isolated from São Francisco Valley (Brazil). |
The text was rewritten (L.391-392) in the manuscript to better describe the resistance mechanisms of plasmids sequenced in this study. |
Reviewer 3 Report
Dear authors, the article is interesting and overall well written. The number of strains is rather limited, but I think the results may be relevant for the scientific community.
Below are some general and specific suggestions that I hope will improve the article:
I recommend standardising the formatting of some words (e.g. One Health; A. veronii). Also, in the first part of the manuscript there are several acronyms (MCR; SFV; TETRA; ANI; NCBI) that should be spelled out in full the first time they are encountered in the text.
In the introduction, the epidemiology of the microorganism could be expanded upon.
In the materials and methods section, it would be useful to report additional information on the isolates (when were they collected? How were they stored? Is an antibiotic sensitivity test available?
Finally, does the study have any limit?
Check comments in the supplementary files.
Regards,
the Reviewer
Author Response
|
Reviewer #3: |
|
|
Comment |
Alteration/Observation |
|
The number of strains is rather limited, but I think the results may be relevant for the scientific community. |
We appreciate your considerations and believe that they will contribute to improving our manuscript. |
|
I recommend standardising the formatting of some words (e.g. One Health; A. veronii). Also, in the first part of the manuscript there are several acronyms (MCR; SFV; TETRA; ANI; NCBI) that should be spelled out in full the first time they are encountered in the text. |
The manuscript was completely revised, and alterations were made to standardize word use. |
|
In the introduction, the epidemiology of the microorganism could be expanded upon. |
We added this information in the introduction. (L.86-103)
|
|
In the materials and methods section, it would be useful to report additional information on the isolates (when were they collected? How were they stored? Is an antibiotic sensitivity test available? |
We included additional information about the isolates in the methods section. (L.480-482). Considering that Aeromonas imposes a challenge to long term maintenance under laboratory conditions, we were not able to perform detailed and consistent phenotypic characterization of these isolates. |
|
Finally, does the study have any limit? |
We can consider two limitations in our manuscript: 1) The lack of a detailed phenotypic characterization of the sequenced isolates and 2) The fact that we were not able to close all sequenced genomes. Despite of this, our study enabled us to gain insight into taxonomic assignment and to better understand virulence and resistance profiles of this species in the One Health context. |
|
Check comments in the supplementary files. |
Unfortunately, no supplementary file was attached to the system. We hope that alterations performed in the revised manuscript could consider these observations. |
Round 2
Reviewer 3 Report
La qualità del manoscritto è stata migliorata sulla base di commenti e suggerimenti.
The quality of the manuscript has been improved based on the comments and suggestions.